# The Use of Digital Social Networks and Engagement in Chilean Wine Industry

**Francisco Egaña [1], Claudia Pezoa-Fuentes [1,\*] and Lisandro Roco [2]**

[1]   Department of Administration, Universidad Católica del Norte, Antofagasta 1240000, Chile; franciscoegab@gmail.com
[2]   Institute of Agricultural Economics, Universidad Austral de Chile, Valdivia 5090000, Chile; lisandro.roco@uach.cl
\*   Correspondence: cpezoa02@ucn.cl; Tel.: +56-552-355767

**Abstract:** Recently, companies and consumers of the wine industry have changed their manner of two-way communication, with the rise of technology that introduces social networks and urges the spread of content. In this study, we identified the use and importance of engagement in social networks such as Facebook (2008 to 2018), Instagram (2012 to 2018) and Twitter (2010 to 2018) since the creation of their official accounts for the main Chilean wineries. The methods used involve qualitative and quantitative approaches that integrate the opinion of a panel of experts to estimate a social media engagement indicator through a descriptive statistical analysis and network analysis, from data originated of 70,856 publications. The results show the upward evolution of engagement, calculated through the interactions seen from users of social networks of the wineries, with users of networks of these wineries leaning towards Facebook in the first place, then Instagram, and Twitter. The contribution of this research lies in the generation of empirical evidence that allows the wine industry in a developing country to enhance its competitive advantage through the correct use of its social networks, the management of its engagement, and the diffusion of new marketing strategies.

**Keywords:** social networks; Chilean vineyards; engagement; business practices; marketing; Chile

## 1. Introduction

Worldwide, the wine industry has evolved rapidly [1], for winemakers, businesspersons and academics, regardless of their geographical area [2–4]. In the same way, information technologies have become relevant for all economic and business sectors [5]. The above is reflected in the importance that the use of social networks is acquiring in companies [6] and how engagement management [7–9], make a difference in a connected and versatile world.

The production and the wine market are part of an internationally prominent economic sector, countries such as France, Italy, Spain and Portugal are world leaders in the wine industry [1]. Countries such as the United States, Australia, Argentina, New Zealand, Chile and China are also part of this list. While most of the studies of this industry are concentrated in developed countries, there are little advances for developing economies [10]. The Chilean wine industry experienced a dynamic growth in recent years [11], the value of Chilean wine exports has increased from US $20 million in the second half of the 1980s to more than US $1400 million on average in the period 2005 to 2007 [12]. However, it is recognized that this industry has been innovative in terms of winemaking and trading [13].

Social media technology allows customers and companies to interact and participate in two-way communication [14,15], in which both the client and company are active participants in the generation and dissemination of content which is generated in an environment that is characterized by a network of people that are interconnected [16]. Participation in social platforms includes how consumers use, share and talk about content related to the brand and company [17]. The first expectation of brands that use social

media marketing is the adaptation of users and contribution to the content and interaction with the brand [18]. Today, most consumers interact with brands through social media, and brands also use social media as customer services and as a fundamental point of contact with consumers [19].

Currently, the use of social networks [20–22] and the interaction of its followers with companies [8], can generate different competitive advantages compared to its competitors [23], establishing that these relationships in social networks can be positive or refusals from their followers or clients [24]. Companies that use their social networks correctly can improve their internal processes, specifically in marketing [25], from this perspective companies tend to have a better relationship with their customers [26].

The objective of the research is to identify the use and importance of engagement in digital social networks (Facebook, Instagram and Twitter: since the creation of their official accounts), through the analysis of 70,856 publications from the main wineries of Chile located in the Metropolitan Region, VI Region of Libertador General Bernardo O'Higgins and VII Region of Maule during the analysis period of a decade from 2008–2018. The methodology used for this study is mixed qualitative and quantitative approach, where it integrates the opinion of experts and subsequently proceeds to analyze the data statistically, implementing the calculation and analysis of engagement, in addition to a network analysis to characterize the behavior of the vineyards based on metrics from Facebook, Instagram and Twitter.

Regarding the literature review, there is evidence of a lack of studies on issues related to [10] in countries of emerging economies [27] and the null study of engagement and social networks in the wine industry. The contribution of this research is the generation of empirical evidence for the wine industry, enhancing its competitive advantage through social networks and engagement.

This article is structured as detailed below: starting with Section 2 that addresses the literature review of social networks, engagement and the wine industry, to continue with section three that focuses on the methodology where it is explained how the sampling was conducted, the data collection procedure and respective analysis, the results are continued, thus ending with the presentation of the discussion, the conclusions and future lines of research.

## 2. Literature Review

### 2.1. Social Networks and Engagement

Social networks exacerbate the change in the ways in which content is produced and consumed online, as they generate an unprecedented abundance of content [5,22], and new forms of interaction with the media, more participatory than those existing in traditional media [28]. Since brands go where consumers are [29], brands have also increased social media marketing in recent years [25]. The use of social media not only comprises a relatively inexpensive communication approach [30], but also opens up new opportunities for brands to extract value from existing and potential consumers, by providing new forms of interaction between brands and such consumers [21,31]. Kumar et al. [26], a well-established fan base can significantly strengthen consumer-brand relationships and has a positive impact on consumer spending.

According to Domene [24], social networks have become the channel of interaction between companies and their customers, the latter use these media to be able to express both positive and negative views of any product or service, share information, generate opinion. This makes the client occupy a fundamental role in the decision-making of companies, when the penetration rate of social networks is investigated, it is observed that the most used social networks in 2020 are Facebook (80%), Instagram (52%) and Twitter (13%) [32]. In the case of Twitter, users can converse using mentions, responses, and hashtags [33]. Despite reports indicating a decline in the popularity and importance of Twitter amid declining investment [34–36] does not report major changes in the percentage of users in the active Twitter accounts. For Instagram it is a mobile application that allows

its users to share photos, take photos, apply filters and share them on the platform itself, as well as on other platforms such as Facebook and Twitter [33].

In the case of Chile, the use of social networks is through desktop computers and smartphones. According to Reuters-Institute [37], 77% of the country's total population has access to the internet, of which 71% use social networks, the most widely used medium to access the content. According to Hootsuite [32], Chilean users of social networks are concentrated on the social network Facebook with 12 million users, Instagram has 8.2 million users, and Twitter 2.47 million users.

Companies must design and develop content that can generate consumer engagement, conversation, and discussion [38]. Engagement each time acquires greater value and is currently a concept worked by companies built through content as well as participation in new platforms (such as social networks). According to the author [39] defines it as "Commitment as the level of motivation of a client, mental state related to the brand and dependent on the context characterized by specific levels of cognition, emotional and behavioral activity in brand interactions" (p. 6). For this research, engagement will be understood as the participation of customers in social networks, this can be passive, through the consumption of the multimedia contents of a publication on social networks, or actively, with the contribution or creation of content related to brands [29,40,41]. Customer engagement has been found to increase loyalty, trust, and branding. Evaluations, which in turn are linked to a brand performance indicator are as important as sales growth [42].

There are studies that try to find attributes or patterns that make a content considered "popular" [43]. In this context, the concept of engagement is established as a measure that quantifies the level of certain forms of "interaction" in social networks [39,44]. This is why consumer engagement is defined as "The intensity of an individuals participation with a brand and the connection with the offers or organizational activities that can be initiated by the client or by the company" (Vivek et al. [45], p. 133). Likes, comments, and shares of brand posts on social media are behavioral manifestations of consumer engagement [46] and fundamental to a brand's overall engagement in social media strategies [31]. e.g., Facebook could help improve the consumer experience while Twitter could improve interactivity [47].

### 2.2. The Wine Industry

The world wine market, according to the the world production 2020 of wine was 253.9 million hectoliters, of which Italy, France, and Spain represented 49% in world and 81% production the European Union. In South America, particularly Argentina and Chile, production decreased due to unfavorable climatic factors. In South Africa production returned to normal production after several years of drought and Australia recorded a low harvest due to forest fires, while New Zealand showed a record harvest volume in 2020. In general, 2020 production is considered below average despite geopolitical tensions, climate change and the COVID-19 pandemic generating a high degree of volatility and uncertainty in the world market for wine [48].

Winegrowing in Chile is a traditional activity that emerged around the 19th century. The revitalization of the land market, the elimination of restrictions on grape plantations, and foreign investment led to the growth of new plantations, the creation of new companies and the modernization of wine-growing facilities. Towards the end of 1980, international wine prices increased and Chilean companies began to grow in number and in technological conditions, stimulating the production of fine and diversified vines [49].

Regarding the Chilean market, according to the [50], the total wine production in 2020 reached 1033 million liters of which the Maule, Libertador Bernardo O'Higgins and Metropolitana regions add to a 99.1% of the country's total production. When analyzing the productions to the types of grape varieties, Cabernet Sauvignon reaches 34.8%, Sauvignon Blanc 14.4%, Merlot 11.8%, Chardonnay with 8.9%, Carmenere 8.6%, and the Syrah variety 6.2%. This is a situation that favors enhancing Chilean exports, taking advantage of using commercial strategies that allow a market positioning for this strain, where in terms

of market share, Chilean wines face very high levels of competition [51]. The Figure 1, shows how the Chilean wine production has evolved in hectoliters for 20 years until 2020. According to [50], a sustained increase can be seen until 2013, where climate change caused production to fluctuate in the last decade noting a decline toward 2020, given the impact of the COVID-19 pandemic and the drought that has affected Chile [48].

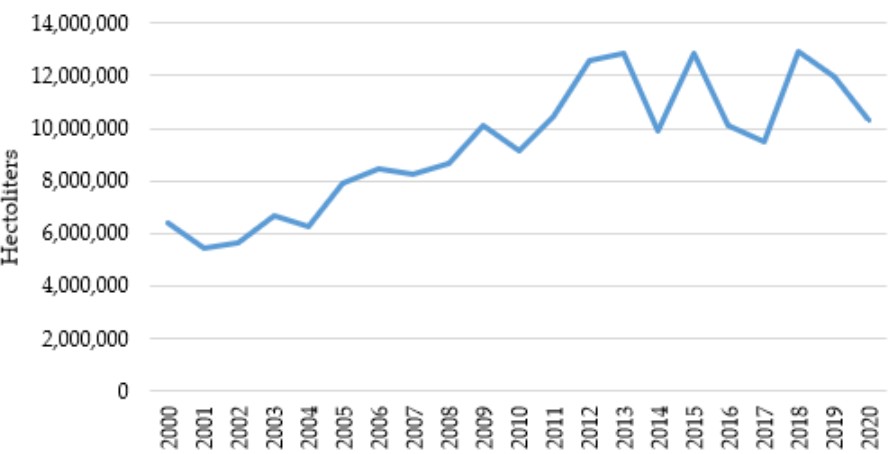

**Figure 1.** Evolution of wine production in Chile (hl).

## 3. Materials and Methods

The method used in this study corresponds to a mixed qualitative and quantitative approach of an exploratory-descriptive type, as the first stage of the research integrates the opinion of many experts, in the field of Chile, Perú, and Spain for the estimation of the weight of engagement, providing a more realistic view of that indicator and, in its second stage, it is based on the calculation of engagement and on the analysis of networks to make visible the formation of a network with a sample of vineyards and their spatial order.

The sampling used for the choice of the vineyards, according to Sampieri et al. [52], is a non-probabilistic or directed type and that "Seeks to specify the properties, characteristics, and profiles of people, groups, communities, processes, objects or any phenomenon that is subjected to an analysis" (p. 80), which is based on general assumptions about the distribution of variables in the population [53]. Furthermore, this sample is of an intentional nature since according to Monje [54] it is possible when "The subjects are chosen intentionally according to criteria established by the researcher, and this selection process is continued practically throughout the entire research process" (p. 46). The wineries that have a presence on the three social networks Facebook, Instagram, and Twitter, are 18: Viña Balduzzi, Viña Bouchon Family, Viña Carmen, Viña Casa Marin, Viña Casas del Bosque, Viña Concha y Toro, Viña Cono Sur, Viña Corral Victoria, Viña De Martino, Viña Gillmore Estate, Viña González Bastías, Viña Leyda, Viña Maipo, Viña Mar de Casablanca, Viña Misiones de Rengo, Viña Montes, Viña Montgras, Viña San Esteban, Viña Santa, Carolina, Viña Santa Helena, Viña Santa Rita, Viña Tarapacá and Viña Undurraga.

Extracted data from social networks allowed obtaining 70,856 publications in total (since the creation of their official accounts) distributed in 21,310 on Facebook (30.1%), 10,771 on Instagram (15.2%) and 38,755 on Twitter (54.1%). As for the data analysis period, it was from 2008 to 2018 and the data collection was carried out from September 2019 to October 2019, manually in the case of Facebook and Instagram and for the social network Twitter the paid web tool Twitonomy was used to download tweets (see Table A1).

### 3.1. Engagement Calculation

To calculate the engagement indicator, 20 experts were invited, both national and international (Chile, Perú, and Spain) from the areas of social media, marketing, agribusiness, strategy, international trade. These experts were sent an online questionnaire in April 2020, and the goal of the questionnaire was to consult the experts on the use and the importance

of social networks by the wine industry. As a result, a response rate of 80% was obtained, equivalent to 16 experts who correctly completed the questionnaire, where the gender of the survey participants was 75% men and 25% women. The evaluation scale with respect to the degree of importance was expressed through an assessment of 1 to 5, with 1 being low importance up to 5 representing the highest degree of importance in what was consulted. After this, a score was calculated based on the weightings made by the experts for the purpose of calculating engagement, Table 1 shows the results of the survey.

**Table 1.** Degree of importance related to the engagement metrics of social networks according to the panel of experts.

| Important Grade | Social Media | Mean | Standard Deviation | Mode | Score [1] |
|---|---|---|---|---|---|
| "Social media for the marketing of the wine industry" | Facebook | 2.75 | 1.00 | 3 | 68.8 |
| | Instagram | 2.81 | 1.05 | 4 | 70.3 |
| | Twitter | 1.81 | 1.11 | 2 | 45.3 |
| "The importance of metrics of engagement on Facebook" | Comments | 2.38 | 0.96 | 3 | 59.4 |
| | Shares | 2.50 | 0.97 | 2 | 62.5 |
| | Likes | 2.50 | 1.03 | 3 | 62.5 |
| "The importance of metrics of engagement on Twitter" | Retweet | 2.19 | 1.11 | 2 | 54.7 |
| | Favorites | 2.00 | 1.11 | 1 | 50.0 |
| | Comments | 2.06 | 1.12 | 1 | 51.6 |
| "The importance of metrics of engagement on Instagram" | Likes | 2.94 | 0.93 | 3 | 73.4 |
| | Comments | 2.82 | 0.98 | 3 | 70.3 |

Note: Score $^1 = \frac{\sum Mark_i * n}{MaxMark * n} * 100$, $n = 16$, $MaxMark = 5$.

The calculation of engagement was made based on [55]. To give more certainty to the calculation of engagement in this research to those metrics that make up the indicator of those authors a weighing from the opinion of the experts was added. In this way the indicator is defined in the following way:

$$Engagement \ in \ Facebook = \sum_{i=l}^{n} (Likes_l * 0.63) + (Comments_l * 0.59) + (Shares_l * 0.63)$$

$$Engagement \ in \ Instagram = \sum_{i=l}^{n} (Likes_l * 0.73) + (Comments_l * 0.7)$$

$$Engagement \ in \ Twitter = \sum_{i=l}^{n} (Retweets_l * 0.55) + (Favorites_l * 0.7) + (Comments_l * 0.52)$$

### 3.2. Network Analysis

The network analysis was carried out through the use of UCINET 6 Software, which is a software package for the analysis of social network data [56]. It allows network analysis, which according to [57] is "The main tool to represent the interactions between individuals or groups of individuals in an illustrative and friendly way" ([57], p. 1). A network is understood as "A group of individuals who interact with others, characterized by the existence of information flows made up of nodes or actors, links and flows" ([57], p. 3). Through the NetDraw tool of the same UCINET 6 software it allows the option of graphing those interactions.

For the authors [58], there was a series of metrics to understand networks and their actors, which helped determine the importance and role of an actor in the network. The most used were classified into centrality and power metrics, and group metrics. Using current metrics and methods, network data could be organized and analyzed to capture the various processes that occurred at different levels of analysis. To establish a deeper assessment of the structure of the graphs, three centrality measures were applied, which were the

most commonly used in the analysis of social networks. The first, "the degree of centrality" (Degree) to the number of links related to a node that allowed it to determine the most important actors of a network with respect to the rest. This number, denominated in degree, is the number of accounts that mentioned or responded to the account in question. It is a simple measure of popularity [58]. The second, "the structural or global centrality" (Eigenvector centrality) aimed to order the network around the notions of "closeness" and "remoteness" according to a natural order between the center, the margin and the periphery of the resulting graph ([59], p. 108). Connecting with someone who is already important in the network will grant more influence than someone who is not important [60]. The third, "the value of intermediation" (Betweenness) expresses the level of influence that a node exerted in the context of the network and the control over the flow of information through all the paths that connected it to other nodes or, in other words, how involved a node was in the relationship structure. According to Yep et al. [60], an account does not have to be popular, or proactive, or have influential friends to be an important member of a network. An account linking two isolated communities could be considered influential as the information travelling between the two groups had to flow through it.

## 4. Results

### 4.1. Main Findings

From the descriptive analysis of the sample of 18 wineries in total, 1,963,416 followers added, of which 82.2% corresponded to the social network Facebook, 13% to Instagram, and 4.9% to Twitter. In Facebook the amount of 1,613,239 (Mean = 86.45; Sd = 190,168.39), in the case of Instagram the amount of 327,981 users (Mean = 13.665; Sd = 18,959.59) and for the social network Twitter 95,153 users (Mean = 95.153; Sd = 4116.25 ), with the social network Facebook being the social platform preferred by users who follow the vineyards. Similarly, a total of 62,113 publications on the three social networks distributed 27.1% on Facebook, 14.5% on Instagram and 58.5% on Twitter. In the case of Facebook, the wineries as a whole had 16,812 publications (Mean = 934; Sd = 514.38), for Instagram 8973 publications were registered (Mean = 498.5; Sd = 401.58) and finally, for Twitter, with 36,328 tweets (Mean = 2018.22; Sd = 1037.24).

For the social network Facebook, the winery with the highest number of followers was identified, which had 824,355 users belonging to Viña Concha y Toro, on the contrary, the winery with the lowest number falls on Viña Santa Cruz with 1391 followers. Regarding the number of publications on this social network, the Montgras Vineyard was the one with the highest number of publications with 1962 and the least frequent with 163 belonging to the Gillmore Estate Vineyard. However, how the group of vineyards mostly manifested itself on Facebook was through publications type "Photos" with 11,915 photographs in total (Mean = 661.94; Sd = 416.5); instead, the publications type "Surveys" was quite small with only 10 surveys in total. In general the publications of the wineries were shared a total of 179,226 times (Mean = 9957; Sd = 14,339.21). This is an important metric since it increased the visibility of the content, which provided an additional advantage for promotions, advertising campaigns and brand recognition, as they could reach consumers around the world, increasing sales [9]. On the other hand, a total of 100,152 publications were commented (Mean = 5564; Sd = 6340.17), which according to [61], comments tend to influence public opinion, since they are among the first things that the users read when they browse, which indirectly affects brand awareness and purchase intent. Regarding the Facebook reactions, officially published in February 2016, they were an extension of the old "Like" button (Period 2004–2016). Its six options (Like, Love, Care, Haha, Wow, Sad and Angry) were represented by lightly edited versions of various emojis that allowed for a more nuanced expression of how users felt towards a post. Users responded to the publications 2,272,827 times in total, the reaction "Like" occupying 88.6% of the reactions (2,013,834 "Likes"), followed by the reaction "Love" with 11% (250,298 "Love") and the valuation "Angry" far below with 0.01% (288 "Angry").

For its part, the social network Instagram Viña Concha y Toro account with 81,000 followers was the most followed in the sample, unlike Viña Maipo with 890 which was the one with the least number of followers. The winery with the highest number of publications is Viña Montes with 1420 publications and, on the contrary, the one with the lowest frequency of publications was Viña Maipo with only 11. Given the three ways of publishing content through Instagram, they were distributed in 88.4% of type "Photos" (Mean = 7929; Sd = 440.1) being the most used, "Videos" with 6.3% (Mean = 31.55; Sd = 29.15) and "Carousel" with 5.3% (Mean = 475; Sd = 26.39). Similarly, the winery with the highest number of comments on the web was Viña Concha y Toro with 7828 comments and, on the contrary, Viña Maipo had only 46 comments.

In the case of the social network Twitter, the winery with the highest number of followers was Viña Concha y Toro with 15,285 followers. On the contrary, the winery with the fewest with only 514 belonged to Viña San Esteban. On the other hand, the winery that tweeted the most was Viña Concha y Toro with 3200 tweets. This figure is due to the maximum limited by the Twitter API, contrasting the above with Viña Tarapacá 425 tweets. About the "Favorites", the vineyard which obtained the greatest amount of this metric was Viña Montes with 7430 favorites, unlike Viña Tarapacá with only 75 of that metric, the lowest of the vineyard sample.

### 4.2. The Engagement

#### 4.2.1. Results the Engagement in Vineyards

The average engagement results, for Facebook had an average of 65.39 (Sd = 82.34), for Instagram 107.07 (Sd = 153.59) and finally Twitter an average engagement of the vineyards of the sample of 1.25 (Sd = 0.55). This is represented in Figure 2, which shows the different levels of engagement of the vineyards studied.

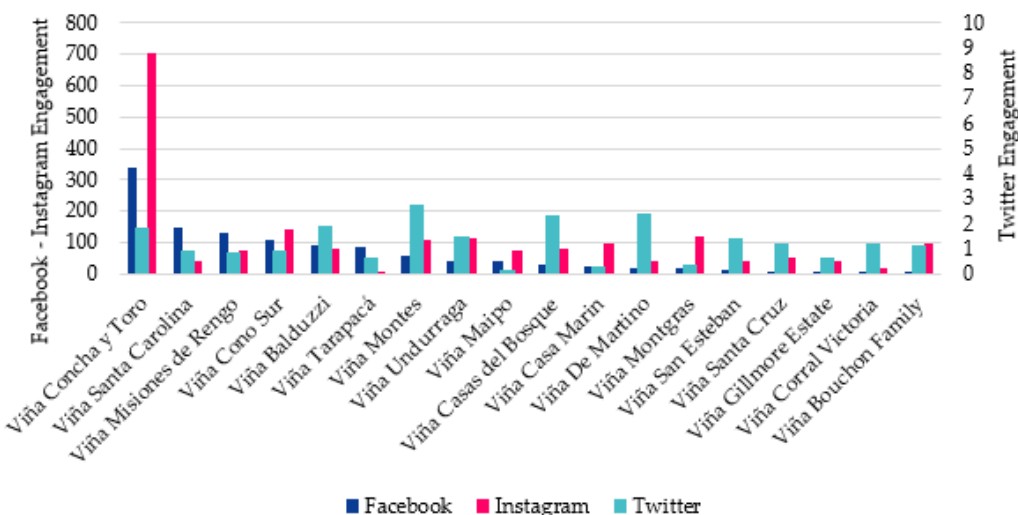

**Figure 2.** Engagement of the vineyards on Facebook, Instagram and Twitter during 2008 to 2018 (*n* = 18).

#### 4.2.2. Engagement in the Different Post of Social Media

For this research it is important to show the way vineyards sought to capture the attention of their customers on social networks. This was how some publications and the different types of these in each social network explained their success in the engagement they managed to obtain from the interaction with their users. Table 2 shows the results obtained in this investigation.

In the case of Facebook, the "compound publications" named for this study occupied the most information or content, as well as photos or videos as a whole and the explanatory text, URLs, or other publications or web pages, or other additional information

(Mean = 76.43; Sd = 122.66) were those that obtained the highest reception by their users. On the contrary, the publications of the "Events" created by the winery qualified at a very low level of the indicator (Mean = 58.45; Sd = 58.65).

In the case of Instagram, we see that photo-type publications were the ones with the highest engagement in vineyard users (Mean = 89.74; Sd = 117.46), contrary to those of the carousel type (Mean = 83.87; Sd = 67.70), where the social network allowed a combined sum of 10 to be combined between photos and/or videos in a single slide-like publication.

Now in Twitter, the common means of communicating information to users was through "tweets". However, these were low in influence (Mean = 0.46; Sd = 0.55) given that "retweets" were of greatest influence (Mean = 5.65; SD = 4.58) when reaching users.

**Table 2.** Engagement in of the vineyards posts (*n* = 18).

| Vineyars | Facebook | | | | | Instagram | | | Twitter | | |
|---|---|---|---|---|---|---|---|---|---|---|---|
| | C. Post | Photos | Videos | Events | Surveys | Photos | Videos | Carousel | Tweets | Retweets | Replies |
| Viña Balduzzi | 29.3 | 96.4 | 14.7 | 0.0 | 0.0 | 76.3 | 95.2 | 83.9 | 0.2 | 4.0 | 2.2 |
| Viña Bouchon Family | 5.0 | 6.1 | 9.4 | 3.3 | 0.0 | 96.5 | 91.8 | 122.2 | 0.8 | 5.4 | 3.6 |
| Viña Casa Marin | 13.0 | 32.4 | 37.1 | 11.7 | 0.0 | 98.1 | 0.0 | 0.0 | 0.2 | 2.9 | 0.7 |
| Viña Casas del Bosque | 47.2 | 27.1 | 18.6 | 0.0 | 0.0 | 75.9 | 66.9 | 125.2 | 0.3 | 2.6 | 2.3 |
| Viña Concha y Toro | 405.2 | 217.1 | 118.8 | 0.0 | 0.0 | 712.9 | 537.0 | 0.0 | 0.3 | 5.4 | 1.9 |
| Viña Cono Sur | 188.8 | 98.3 | 72.9 | 0.0 | 6.1 | 143.7 | 130.2 | 119.5 | 2.2 | 15.4 | 2.0 |
| Viña Corral Victoria | 0.5 | 8.7 | 21.1 | 0.9 | 0.0 | 19.8 | 24.9 | 28.6 | 0.5 | 12.3 | 2.3 |
| Viña De Martino | 18.0 | 19.6 | 36.0 | 0.0 | 0.0 | 42.3 | 47.8 | 71.3 | 0.3 | 3.4 | 2.4 |
| Viña Gillmore Estate | 4.8 | 9.0 | 2.5 | 0.0 | 0.0 | 39.2 | 40.3 | 0.0 | 0.1 | 14.4 | 4.1 |
| Viña Maipo | 37.5 | 41.4 | 10.0 | 0.0 | 0.2 | 46.0 | 91.6 | 179.8 | 0.1 | 1.3 | 0.2 |
| Viña Misiones de Rengo | 366.8 | 88.0 | 259.0 | 0.0 | 0.6 | 72.4 | 68.8 | 115.5 | 0.7 | 2.3 | 0.9 |
| Viña Montes | 53.3 | 61.4 | 36.7 | 3.1 | 0.0 | 109.1 | 109.8 | 235.2 | 1.4 | 4.8 | 3.3 |
| Viña Montgras | 7.4 | 19.9 | 17.3 | 5.6 | 0.0 | 118.1 | 105.3 | 155.5 | 0.0 | 1.6 | 0.5 |
| Viña San Esteban | 5.7 | 14.2 | 20.2 | 8.6 | 0.0 | 41.4 | 37.9 | 41.6 | 0.4 | 4.7 | 2.1 |
| Viña Santa Carolina | 129.2 | 159.3 | 61.2 | 0.0 | 382.9 | 43.0 | 48.6 | 79.2 | 0.2 | 2.8 | 1.6 |
| Viña Santa Cruz | 5.7 | 9.1 | 12.4 | 0.0 | 0.0 | 2.6 | 0.0 | 0.0 | 0.4 | 12.3 | 5.9 |
| Viña Tarapacá | 20.0 | 102.7 | 44.6 | 0.0 | 0.0 | 37.0 | 43.5 | 44.7 | 0.1 | 1.9 | 0.8 |
| Viña Undurraga | 38.4 | 41.5 | 28.9 | 0.0 | 0.0 | 120.7 | 75.8 | 107.8 | 0.2 | 4.3 | 2.2 |

It is interesting to understand how the engagement in aggregate of the vineyards is distributed in the social networks of this research (see Figure 3). We see that in the case of Facebook the type of compound publication covered 37% of the engagement, which stood out for its high level of associated content to publication and, on the contrary, events and surveys did not stand out for capturing the attention of their users. The case of Instagram was the same way, with 38% of the photo-type publications being the ones with the highest engagement. However, the three types of publications had similar behaviors when reaching their users. Finally, in the case of Twitter, it was the retweets that had the greatest arrival on this social network. Note the fact that, although the tweet was the standard means to communicate through this social network, it is striking that it had a much lower level of engagement compared to comments and retweet type publications with 68%, thus showing the effectiveness of content generation in this social network when they have a high level when sharing content.

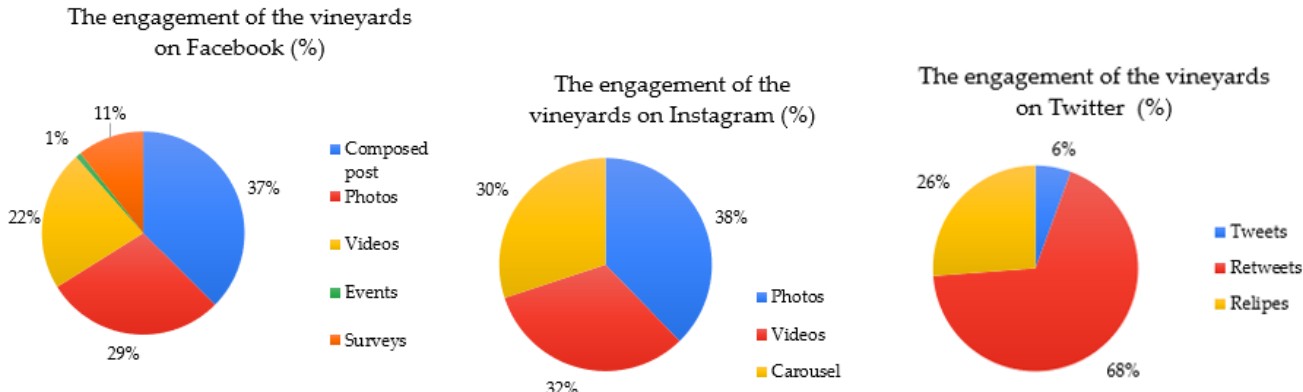

**Figure 3.** Distribution of the engagement of the publications in the social media of the vineyards.

### 4.2.3. Variation in the Engagement of Vineyards on Social Media between 2008 and 2018

The behavior of the vineyards in the period studied was not the same in the three social networks as evidenced in the Figure 4. Reviewing the temporal evolution of engagement, it is clear that the vineyards had begun to mark their presence on social networks. Compared to the other two, Facebook was the social network which the sample of vineyards occupied from the first year of the study period, although at the beginning with few publications given the low popularity of the social network in those years, until 2017 a sustained growth in its use. In the case of Instagram, it was very significant for this research, since the popularization of this social network at the end of 2011, the use of this social network as a means of interaction with users was of sustained exponential growth until the last year of the present investigation. In the case of Twitter, although engagement was irregular in terms of growth, it is evident that in the last years of the research, popularity among vineyard followers decreased, either due to low vineyard-user interaction or due to the lack of interest of the vineyards in publishing striking content for their followers.

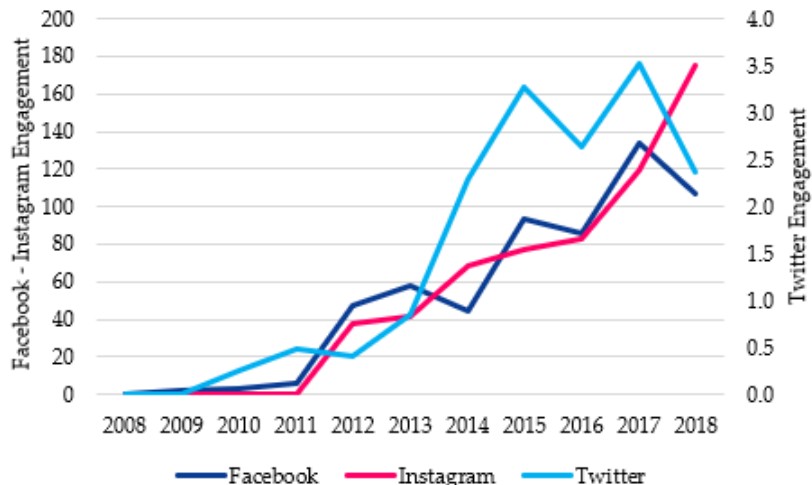

**Figure 4.** Distribution of engagement in the social networks of the vineyars as a whole during the period studied.

### 4.2.4. Correlation between Engagement and Social Media Metrics

The Pearson Correlation was used to identify the relation between the engagement metrics and the social media metrics of the sample. In the case of Facebook (Table 3), regarding to the metrics used, all the above showed a correlation coefficient greater than 0.5. Firstly, between engagement and reactions ($r = 0.977$; $p < 0.001$), secondly between engagement and the number of times users shared content ($r = 0.961$; $p < 0.001$), and thirdly,

engagement and the number of users who followed the winery on Facebook (r = 0.909; *p* < 0.001) and in fourth place the relation between engagement and comments on the winery's posts (r = 0.857; *p* < 0.00). Contrary to the above, note that engagement did not have a substantial relation between the indicator and the number of publications made by the winery, thus being a medium correlation (r = 0.477; *p* < 0.045). It can be seen that engagement on Facebook did not depend to a great extent on the amount or intensity with which the winery exposed its content on the social network, but rather on the interactions expressed in Facebook reactions, particularly through the button "Like" the ones that explained the greater engagement of the winery.

**Table 3.** Pearson correlations of the metrics on Facebook (*n* = 18).

|  | Followers | Post | Reactions | Comments | Shares | Engagement |
|---|---|---|---|---|---|---|
| Followers | 1 |  |  |  |  |  |
| Post | 0.358 | 1 |  |  |  |  |
| Reactions | 0.895 ** | 0.554 * | 1 |  |  |  |
| Comments | 0.721 ** | 0.559 * | 0.834 ** | 1 |  |  |
| Shares | 0.941 ** | 0.506* | 0.958 ** | 0.863 ** | 1 |  |
| Engagement | 0.909 ** | 0.477 * | 0.977 ** | 0.857 ** | 0.961 ** | 1 |

Note: * *p* < 0.05, ** *p* < 0.01.

In the case of Instagram (Table 4), the metric which was most significantly correlated, were the "Likes" (r = 0.897; *p* < 0.001). Followed by this was the significant relationship between engagement and the number of followers of the winery (r = 0.881; *p* < 0.001) and also, the relationship between engagement and user comments on the winery's publications in the social network (r = 0.795; *p* < 0.001). Contrasting the above, there was a negligible correlation between the number of publications and the engagement generated in the social network (r = 0.044, *p* < 0.862), thus showing that the number of publications did not cause a given level in the users of the social network of engagement, but rather the quality of the content—expressed by its users through the positive evaluation of the "Likes" grants a higher level of engagement.

**Table 4.** Pearson correlations of the metrics on Instagram (*n* = 18).

|  | Followers | Post | Likes | Comments | Engagement |
|---|---|---|---|---|---|
| Followers | 1 |  |  |  |  |
| Post | 0.334 | 1 |  |  |  |
| Likes | 0.908 ** | 0.453 * | 1 |  |  |
| Comments | 0.883 ** | 0.529 * | 0.941 ** | 1 |  |
| Engagement | 0.881 ** | 0.044 * | 0.897 ** | 0.795 ** | 1 |

Note: * *p* < 0.05, ** *p* < 0.01.

Regarding the social network Twitter (Table 5), a significant correlation stood out in three of the five metrics with which the analysis was carried out. In the first place, between the engagement of the vineyards and the favorites (r = 0.765; *p* < 0.001) it obtained the highest correlation. Continuing like this, a medium correlation between engagement and the number of followers (r = 0.462; *p* < 0.053). However, the number of tweets (r = 0.153; *p* < 0.545), comments (r = 0.042; *p* < 0.869) and mentions (r = 0.01; *p* < 0.97) had an insignificant correlation to achieve presence in the social network Twitter. This is why in the case of this social network, the evaluations by users towards the publications of the vineyards on the network (tweets or retweets) were important along with the number of followers.

**Table 5.** Pearson correlations of the metrics on Twitter ($n = 18$).

|            | Tweets    | Followers | Relipes | Favorites | Mentions | Engagement |
|------------|-----------|-----------|---------|-----------|----------|------------|
| Tweets     | 1         |           |         |           |          |            |
| Followers  | 0.757 **  | 1         |         |           |          |            |
| Relipes    | 0.674 **  | 0.292     | 1       |           |          |            |
| Favorites  | 0.525 *   | 0.550 *   | 0.382   | 1         |          |            |
| Mentions   | −0.127    | −0.045    | 0.197   | −0.056    | 1        |            |
| Engagement | 0.153     | 0.462     | 0.042   | 0.765 **  | 0.01     | 1          |

Note: * $p < 0.05$, ** $p < 0.01$.

### 4.3. Network Analysis

In the case of Facebook, 16,812 publications, 1,613,239 followers, 2,094,748 Facebook reactions (Like, Love, Care, Haha, Wow, Sad and Angry), 100,152 comments, and 179,226 shares were analyzed.

Following this, note that the analysis discriminated between five vineyards, which were Viña Bouchon Family, Viña Corral Victoria, Viña Gillmore Estate, Viña San Esteban, and Viña Santa Cruz, resulting in a network of vineyards of 13 nodes. This network had a mean geodetic distance of 1.615 and a standard deviation of 0.487. In the same way, the network with 13 actors had a network density of 19.6% which, based on the scale proposed by Coronado-Padilla [62] for the density´s measurement, could be determined as a low density, where it showed that the network was not very cohesive, where according to Chandes and Paché [63], Maghsoudi and Pazirandeh [64], Tatham and Spens [65] it ratifies a problem in the coordination between the actors.

When the centrality metrics were analyzed, note that the average degree of the network was 3.333 degrees. Note that Viña Concha y Toro had 12.00 degrees, the highest in the network, positioning the vineyard as the most influential within the group. The next most influential vineyard in the network was Viña Santa Carolina obtaining 10.00 degrees, where together with the first vineyard, they were positioned in the central part of the network. It is equally important to show that the De Martino and Casa Marin vineyards had a fairly low influence, both only with 1.0 degrees where both were only related to the two previously highlighted vineyards. Similarly, analyzing the structural centrality, Viña Concha y Toro obtained the highest value with 0.44, evidencing the influence that it had as a central node in the network, thus highlighting the importance of this vineyard in the sample as a channel and the fundamental core of the network, however, there was also Viña Santa Carolina with 0.42 which, as a whole, was the link between the group of vineyards and those that were not connected to the others as axes of the network. Equally important, note that a group of five vineyards was formed, which shared similar characteristics as shown in Figure 5 on the right side of the network.

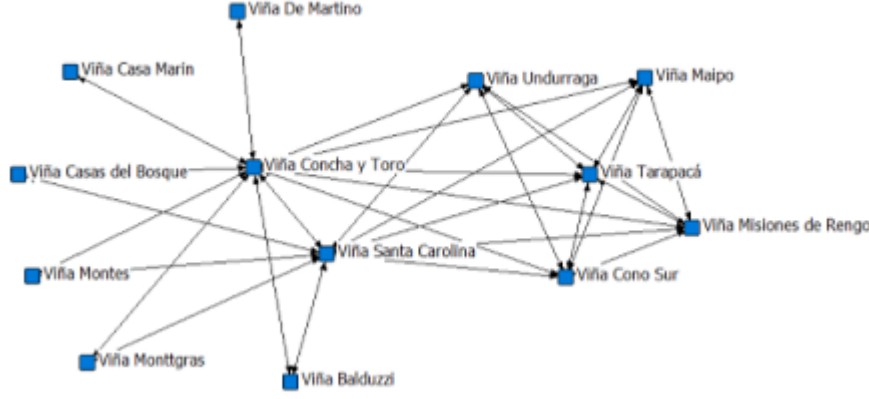

**Figure 5.** Graphic network of vineyards on Facebook during 2008 to 2018 ($n = 13$).

Regarding the measurement of the intermediation value, in the same way as the previous measures, Viña Concha y Toro was found with the highest degree of intermediation with 34.2, supporting the importance that this vineyard played in the network on the social network Facebook.

For the social network Instagram, we analyzed 8973 publications, 255,024 followers, 1,401,255 "likes" and 31,602 comments. The procedure did not consider Viña Balduzzi, Viña Bouchon Family, Viña Casa Marín, Viña Corral Victoria, and Viña Maipo from the graphic network.

This network had a mean geodetic distance of 1.564 and a standard deviation of 0.496. In the same way, the network with 13 actors had a network density of 0.22 which, based on the scale proposed by [62] for measuring density, it can be determined that the network had a low density, where it showed that the network was not very cohesive and where according to [63–65] it ratified a problem with the coordination between the actors.

Regarding the centrality metrics, it is evident that the average degree of the network was 3.778. Note that Viña Concha y Toro had 12.00 degrees, the highest in the network, positioning the vineyard as the most prominent influential within the group, as in the case of Facebook. The next most influential vineyard in the network was Viña Montes obtaining 9.0 degrees, where together with the first vineyard, they were positioned in the central part of the network, also note that Viña Montes belonged to an affiliated group of vineyards on the left side of the network, as shown in Figure 6, a group of seven vineyards which shared similar characteristics, giving Viña Concha y Toro the centrality of the network. Contrasting the above, the low representativeness that Viña San Esteban (0.11) and Viña Santa Carolina (0.167) had in the sample, the latter showed a substantial difference in this case on Instagram to Facebook.

Now, regarding to structural centrality, Viña Concha y Toro obtained the highest value with 0.4, demonstrating the influence in the central section of the network, very closely Viña Montes was found with 0.38. Continuing with the measures of centrality, the value of intermediation, Viña Concha y Toro has the highest degree of intermediation with 35.57, as in Facebook, thus showing the relevance of this vineyard in the sample of this research.

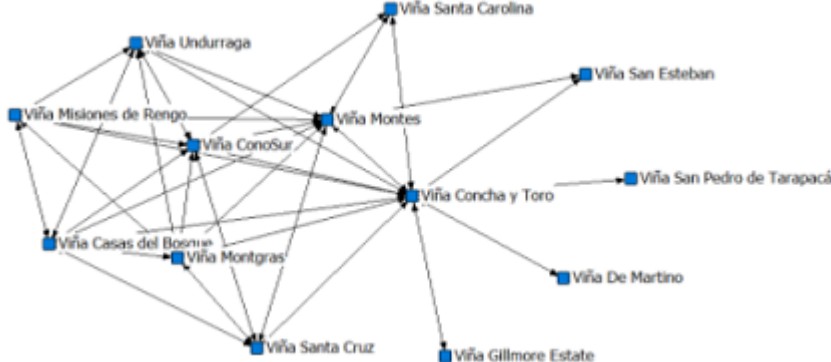

**Figure 6.** Graphic network of vineyards on Instagram during 2008 to 2018 (*n* = 13).

In the case of Twitter, the analysis consisted of 36,328 tweets, 95,153 followers, 3057 comments, 31,301 favorites, and 609 mentions. The procedure for creating the graphic network left out Viña Bouchon Family, Viña San Esteban, and Viña Undurraga. This network had a mean geodetic distance of 1.533 and a standard deviation of 0.499. In the same way, the network with 15 actors had a network density of 0.32 which, based on the scale proposed by [62] for measuring density, it can be determined that the network had a low density, where it showed that the network was not very cohesive, which according to [63–65] ratified an issue of coordination between the actors.

Regarding the centrality metrics, it is evident that the average degree of the network was 5.444. Regarding the vineyard, which had the highest degree Viña Concha y Toro has 14.00 degrees, thus being the most important vineyard within the group. Followed by this

vineyard was Viña Montes, the same phenomenon as on Instagram, obtaining 13.0 degrees. It should also be noted that a group of seven vineyards was formed the right part of the network, that share similar characteristics as shown in Figure 7, highlighting the centrality of Viña Concha y Toro network. However, it shows the low representativeness that Viña Tarapacá had in the sample with only 1.0 degrees, having no interaction with the other vineyards. Continuing with the analysis, referring to structural centrality, Viña Concha y Toro obtained the highest value with 0.368, demonstrating representativeness in the central section of the vineyard network and in a similar way, Viña Montes with 0.384. Regarding the value of intermediation, Viña Concha y Toro had the highest degree of intermediation with 32.94, as in Facebook and Instagram, showing the role that this vineyard had for the interconnection between the rest of the vineyards and the preponderance that it had in the Chilean national wine market.

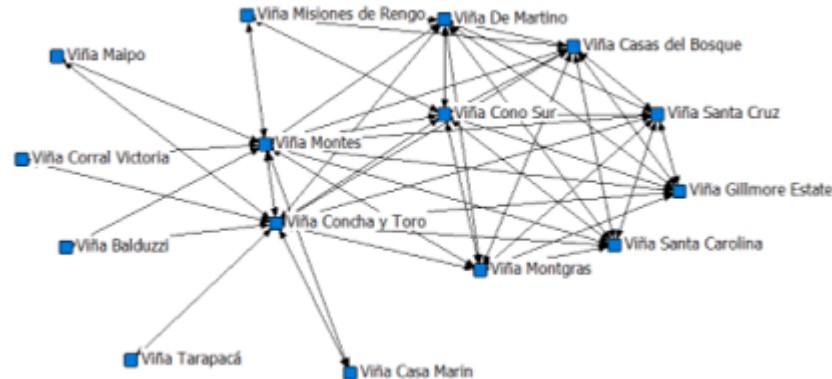

**Figure 7.** Graphic network of vineyards on Twitter during 2008 to 2018 (*n* = 15).

## 5. Discussion

This article analyzes the use of social media networks Facebook, Instagram and Twitter of the main wineries in Chile (since the official creation of their account), by using an engagement indicator generated with their users.

The authors [66], in their international study of the use of social networks, corresponding to the analysis of vineyards from Germany, the USA, New Zealand, and Australia, they conclude that Facebook is their main interaction network, as in this investigation as shown in Figure 2, through the engagement that Chilean wineries generate with their users.

The results of Figure 1 clearly show the growth experienced by wine production in Chile from the beginning of 2000 to 2013, where from that year [50] indicates the decrease in production due to contingencies in which is the country involved, such as the social, economic and health environment. In Table 2 and Figure 3 of this study identifies the use of social networks by the vineyards as indicated [5], from the exchange of information and use new interactions with the media as confirmed [28].

In the results of Tables 3–5 of the research, it is visualized that brands are where they have the greatest number of consumers, like the study of [29], confirming this study that the use of social networks they do not have a single economic approach as for [30], having new opportunities for the brand, such as the works of [21,31].

For the study by [33], it was determined that users publish on Instagram, Facebook and Twitter, they are photos and filters applied to the photos, on the other hand in this study and as evidenced in Figure 2 the majority of data that the users of the Chilean vineyard accounts publish on Twitter 68% in Tweets, for Facebook 37% in publications and Instagram have the majority with 38% in publishing photos. This is similar to what [67] that shows the use of social networks by Sicilian vineyards on Facebook that exhibited a high level of informative content, especially photos, and interaction with users. The authors also argue that small businesses have been more concerned with content intensity and responsiveness with their users.

This study, as in [24], confirms that social networks are a channel of interaction between clients and companies. It is essential to enhance the role played by information technologies, such as the internet and social networks, and how they can favor the engagement of companies and improve their results, as has been evidenced by [68] and the lack of cooperative work and the correct interaction of spatial and non-spatial proximity [69].

According to the authors [4] highlight the importance of cooperation and social proximity to enhance and obtain better results in marketing issues in agribusiness, evidenced in Figures 5–7 of the network analysis of this investigation.

This study shows that the proper management of social networks and the use of new technologies, as in [10], enhance the growth, differentiation and innovation of companies in the wine industry.

In reference to the engagement obtained from the vineyards, the vineyard with the highest engagement is Viña Concha y Toro for Instagram and Facebook, in the case of Twitter it is for Viña Montes. We can conclude that the reactions (likes) of users are the most important thing in order to achieve a positive engagement on behalf of the Chilean wineries since it is highly correlated with the three social networks, which allow users to express their feelings in front of the brand without words [70], these aspects are important since a large number of "likes" can transmit a signal that affirms the positivity and importance of the publication [71].

## 6. Conclusions

This research analyzed the use of the social networks Facebook, Instagram and Twitter of the main wineries in Chile for a decade (since the official creation of their account), analyzing 70,856 publications, through the engagement indicator generated with their users.

The results reveal the importance of social network users through interactions, through this research it is possible to identify which is the interaction with the greatest impact for each social network. For Facebook, the greatest interaction is achieved through its "compound publications", which have a greater volume of information and photos, for Instagram it is through photos and videos and for Twitter it is through its retweets, showing the fundamental role in promoting the vineyards. However, there is a significant finding when comparing the engagement indicator and the size of the company (large companies or SMEs), since, regardless of the marketing efforts of companies through social networks, they do not ensure a level of engagement reflected in this research, for example, nationally and internationally recognized wineries do not have the high levels of engagement expected with their users.

Regarding the variation of engagement in the decade of study, it was evidenced that in the first years until 2017 the three social networks increased steadily, but from that year on, a particular phenomenon was identified with the social networks Facebook and Twitter lost its popularity among its users for the last years of the investigation. However, an important discovery is the exponential increase from 2011 to 2018 of the social network Instagram, demonstrating the importance of the relationship with its users through this social network for the Chilean wine industry.

The formation of networks based on the sample of vineyards shows the representativeness of Viña Concha y Toro in this research, since the results show that this vineyard is a central axis among the groupings of vineyards in each social network as a communicating axis between vineyards that share close attributes and with the vineyards without transcendence in the network.

The limitations of this study lie in the cross-section of the data and because the sample analyzes the main Chilean wineries in a period of a decade, and not the vast majority existing in Chile or distinguishing the size of the company, which the results cannot be generalized, giving way to new research in the context of the wine industry and agribusiness, national and international, and in the same way for other types of industries, even developing in-depth studies with each social network separately. We believe that in future research it is important to incorporate new metrics for calculating engagement

to provide a deeper analysis and new statistical analysis for the study of specific areas of engagement in social networks. It is also interesting to the study of engagement, in contexts outside normality, so it could be applied in the current health context (COVID-19) to identify the similarities and differences in the social networks of this wine industry, highlighting the increase in electronic commerce [72,73]. However, the application of this indicator is applied in the social networks of any industry to make comparisons between companies in different industries.

**Author Contributions:** Conceptualization, L.R., F.E. and C.P.-F.; methodology, F.E., C.P.-F. and L.R.; validation, F.E. and L.R.; formal analysis, F.E.; investigation, F.E., L.R. and C.P.-F.; writing—original draft preparation, F.E.; writing—review and editing, F.E., L.R. and C.P.-F.; visualization, F.E.; supervision, C.P.-F. All authors have read and agreed to the published version of the manuscript.

**Funding:** This research received no external funding.

**Data Availability Statement:** Data are available upon request.

**Acknowledgments:** The authors thank to Samuel Lutino and Javier Barreda for their collaboration in database construction. Also, thanks to the two anonymous referees for its suggestions that improve the work.

**Conflicts of Interest:** The authors declare no conflict of interest.

## Appendix A

**Table A1.** Start date of the collection of publications from the vineyards social networks.

| Vineyards | Facebook | Instagram | Twitter |
| --- | --- | --- | --- |
| Viña Balduzzi | 31-07-2012 | 14-12-2015 | 16-10-2012 |
| Viña Bouchon Family | 08-09-2014 | 27-10-2014 | 03-09-2014 |
| Viña Casa Marin | 06-09-2011 | 01-10-2015 | 13-01-2010 |
| Viña Casas del Bosque | 04-07-2015 | 14-06-2013 | 30-06-2011 |
| Viña Concha y Toro | 10-07-2011 | 08-08-2012 | 30-01-2017 |
| Viña Cono Sur | 04-02-2013 | 05-02-2013 | 31-08-2018 |
| Viña Corral Victoria | 27-11-2015 | 01-07-2017 | 26-03-2014 |
| Viña De Martino | 15-10-2010 | 07-12-2012 | 09-03-2010 |
| Viña Gillmore Estate | 12-06-2011 | 30-08-2015 | 16-03-2010 |
| Viña Maipo | 02-04-2012 | 15-07-2016 | 29-08-2011 |
| Viña Misiones de Rengo | 07-05-2008 | 25-08-2014 | 14-02-2015 |
| Viña Montes | 06-12-2016 | 03-10-2012 | 13-02-2015 |
| Viña Montgras | 19-07-2010 | 12-04-2016 | 02-07-2010 |
| Viña San Esteban | 20-08-2009 | 02-03-2016 | 26-03-2012 |
| Viña Santa Carolina | 03-06-2009 | 11-03-2014 | 12-05-2010 |
| Viña Santa Cruz | 23-04-2017 | 19-12-2015 | 30-07-2013 |
| Viña Tarapaca | 12-07-2012 | 09-12-2016 | 13-10-2011 |
| Viña Undurraga | 01-01-2015 | 28-09-2015 | 17-08-2010 |

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
