# Peer review of "The Use of Digital Social Networks and Engagement in Chilean Wine Industry"

_jtaer, doi:10.3390/jtaer16050070_

Round 1
Reviewer 1 Report
The article treats a very interesting theme, but not only because we are in a pandemic time and the social networks play a very important role in buying/selling online, the authors must update the information used in the literature review.
For example in 1.1. Social networks, the data must be at least from 2020, not fro, 2017 („it is observed that the most used social networks in 2017 are Facebook (63%), Instagram (27%) and Twitter (22%) [31]”). Thus, in order to have the last info about the analysed field, the statistics must be up-to-date.
The same thing must be done for the following statistic, which is from 2018: „According to the [36], 77% of the country’s total population has access to the internet, of which 71% use social networks, the most widely used medium to access the content” and may be updated using official data from the statistics presented on www.worldinternetstats.
Another example is for 1.2. The wine industry: „The world wine market, according to the [48] the world production of wine was 260 million hectoliters, of which Italy, France, and Spain represent 48%. The statistics is from 2018, an up-to-date is required, for 2020 at least. Maybe a comparison is better, for example the authors may offer data from 2015, 2018 (to use the data already offered) and from 2020, to observe the trend. A graphical representation will offer a better visualization for these statistics and will improve the content of the article.
A graphical representation is required also for: „Regarding the Chilean market, according to the [26], the total wine production in 2019 reached 1,200 million liters”, in order to observe the trend in the past, present, and according to the regression function used in the research to forecast a future wine production (a table from 2010-2020 will be better for total wine production).
I noticed a good research and a good idea used for analysis.
A final request is about adding a few books, almost the entire article is only using other article from journals, and the books could be used from e-commerce, e-communication, social networks, mobile apps, books from 2020-2021.
Thank you and good luck!

Author Response
Referee 1
According to your instructions the following changes are made
- The article treats a very interesting theme, but not only because we are in a pandemic time and the social networks play a very important role in buying/selling online, the authors must update the information used in the literature review.
Literature update is performed
Aliste, E.; Bustos, B.; Gac, D.; Schirmer, R. Discursos sobre la viña y el vino: nuevos territorios en el imaginario social. Revista de geografía Norte Grande 2019, 72. doi:502http://dx.doi.org/10.4067/S0718-34022019000100113.50312.
Cusmano, L.; Morrison, A.; Rabellotti, R. Catching up Trajectories in the Wine Sector: A Comparative Study of Chile, Italy, and South Africa. World Development 2010, 38, p1588–1602. doi:https://doi.org/10.1016/j.worlddev.2010.05.002.50613.
Varas, M.; Basso, F.; Maturana, S.; Pezoa, R.; Weyler, M. Measuring efficiency in the Chilean wine industry: a robust DEA approach. Applied Economics 2021,53, p1092–1111. doi:50810.1080/00036846.2020.1826400.
- For example in 1. Social networks, the data must be at least from 2020, not fro, 2017 („it is observed that the most used social networks in 2017 are Facebook (63%), Instagram (27%) and Twitter (22%) [31]”). Thus, in order to have the last info about the analysed field, the statistics must be up-to-date.
is replaced by
This makes the client occupy a fundamental role in the decision-making of85companies, when the penetration rate of social networks is investigated, it is observed86that the most used social networks in 2020 are Facebook (80%), Instagram (52%) and87Twitter (13%) [30].
- The same thing must be done for the following statistic, which is from 2018: „According to the [36], 77% of the country’s total population has access to the internet, of which 71% use social networks, the most widely used medium to access the content” and may be updated using official data from the statistics presented on worldinternetstats.
is replaced by
According to the [35], 77% of the country’s total population has access to95the internet, of which 71% use social networks, the most widely used medium to access96the content. According to [30], Chilean users of social networks are concentrated on97the social network Facebook with 12 million users, Instagram has 8.2 million users, and98Twitter 2.47 million users
- Another example is for 2. The wine industry: „The world wine market, according to the [48] the world production of wine was 260 million hectoliters, of which Italy, France, and Spain represent 48%. The statistics is from 2018, an up-to-date is required, for 2020 at least. Maybe a comparison is better, for example the authors may offer data from 2015, 2018 (to use the data already offered) and from 2020, to observe the trend. A graphical representation will offer a better visualization for these statistics and will improve the content of the article.
is replaced by
The production and the wine market are part of an internationally prominent23economic sector, countries such as France, Italy, Spain and Portugal are world leaders in24the wine industry [1]. Countries such as the United States, Australia, Argentina, New25Zealand, Chile and China are also part of this list. While most of the studies of this26industry are concentrated in developed countries, there are little advances for developing27economies [10]. The Chilean wine industry experienced a dynamic growth in recent28years [11], the value of Chilean wine exports has increased from US$20 million in the29second half of the 1980s to more than US$ 1,400 million on average in the period 2005 to302007 [12]. However, it is recognized that this industry has been innovative in terms of31winemaking and trading [13].
- A graphical representation is required also for: „Regarding the Chilean market, according to the [26], the total wine production in 2019 reached 1,200 million liters”, in order to observe the trend in the past, present, and according to the regression function used in the research to forecast a future wine production (a table from 2010-2020 will be better for total wine production).
Regarding the Chilean market, the total wine production in1412020 reached 1,033 million liters of which the Maule, Libertador Bernardo O’Higgins142and Metropolitana add to a 99.1% of the country’s total production.143When analyzing the productions to the types of grape varieties, Cabernet Sauvignon144reaches 34.8%, Sauvignon Blanc 14.4%, Merlot 11.8%, Chardonnay with 8.9%, Carmenere1458.6%, and the Syrah variety 6.2%, which favor Chile by enhancing 146exports, taking advantage of using commercial strategies that allow a market positioning147for this strain, where in terms of market share, Chilean wines face high levels of148competition [49]. The figure below shows how the Chilean wine production has evolved149in hectoliters for twenty years until 2020. According to [48], a sustained increase can150be seen until 2013, where climate change has caused production to fluctuate in the last151decade noting a decline toward 2020, given the impact of the COVID-19 pandemic and152the drought that has affected Chile [46].
I noticed a good research and a good idea used for analysis.
- A final request is about adding a few books, almost the entire article is only using other article from journals, and the books could be used from e-commerce, e-communication, social networks, mobile apps, books from 2020-2021.
e-commerce, Numerical location in the references is : number 73
Cohan, P. Goliath Strikes Back: How Traditional Retailers Are Winning Back Customers from Ecommerce Startups, 1st ed. ed.; Apress, 2020.
e-communication, Numerical location in the references is : number 15
Belch, G.; Belch, M.; Kerr, G.; Waller, D.; Powell, I. Advertising: An Integrated Marketing Communication Perspective, 4 ed.; McGraw-Hill Connect, 2020.
Social networks, Numerical location in the references is : number 22
Kane, A. Social media marketing and online business 2021; 2020.

Reviewer 2 Report
This manuscript concerns an interesting topic, but it is poorly organized and structured. First of all, the title needs to be reorganized. The Abstract section is overgeneralized. For example, line 1, “The wine industry generates great global interest”. This sentence has no practical meaning, failing to clarify the background and problems of this research. It is recommended to rearrange the structure of this manuscript in the order of Introduction, Literature review, Material and method, Results, Discussion, Conclusions, in a bid to increase the logic and readability of the manuscript.
In the Introduction section, the clarifications of the necessity of this research and the problems to be studied are seriously insufficient, and the relevance between the digital social network and the Chilean wine industry shall be explained. In addition, the research time range is set to be 2008-2018, so what is the necessity of this research?
The citation in Line44 shall follow the MDPI style.
In Line 58-62, the manuscript has insufficient supporting evidence for the lack of relevant research. Please reconfirm the structure in Line 63-68.
In Line 69-143, there are structural problems, and attention shall be given to the consistency of the article structure. For example, it only has 1.1.1 Engagement.
In the Materials and Methods section, there are differences in the sampling ratios for Facebook, Instagram, and Twitter—in particular, the sampling ratio for Twitter exceeding 50% of the total sample size. What’s more, to my knowledge, Instagram was launched in 2010, but the data offered was from 2008 to 2018, which made me concerned about the accuracy of data analysis.
The Results and Conclusions sections are poorly organized. It is recommended to rewrite the Results section and add a Discussion section. Furthermore, the conclusion is not robust enough, and I would like to see the differences between this research and previous research as well as its innovations.
Author Response
Referee 2
According to your instructions the following changes are made
- This manuscript concerns an interesting topic, but it is poorly organized and structured. First of all, the title needs to be reorganized.
The title has been replaced by: The use of digital social networks and engagement in Chilean wine industry.
- The Abstract section is overgeneralized. For example, line 1, “The wine industry generates great global interest”. This sentence has no practical meaning, failing to clarify the background and problems of this research.
The abstract has been replaced by:
Recently, companies and consumers of the wine industry have changed their1manner of two-way communication, with the rise of technology that introduces social networks2and urges the spread of content. In this study, we identified the use and importance of engagement3in social networks such as Facebook (2008 to 2018), Instagram (2012 to 2018) and Twitter (20104to 2018) since the creation of their official accounts for the main Chilean wineries. The methods5used involve qualitative and quantitative approaches that integrate the opinion of a panel of6experts to estimate a social media engagement indicator through a descriptive statistical analysis7and network analysis, from data originated of 70,856 publications. The results show the upward8evolution of engagement, calculated through the interactions seen from users of social networks9of the wineries, with users of networks of these wineries leaning toward Facebook in the first10place, then Instagram, and Twitter. The contribution of this research lies in the generation of empirical evidence that allows the wine industry in developing countries to enhance its competitive12advantage through the correct use of its social networks, the management of its engagement, and13the diffusion of new marketing strategies.
- It is recommended to rearrange the structure of this manuscript in the order of Introduction, Literature review, Material and method, Results, Discussion, Conclusions, in a bid to increase the logic and readability of the manuscript.
The manuscript has been restructured: Introduction, Literature review, Material and method, Results, Discussion, Conclusions. Accordingly, with the Introduction rewritten and the Discussion and Conclusions added separately.
- In the Introduction section, the clarifications of the necessity of this research and the problems to be studied are seriously insufficient, and the relevance between the digital social network and the Chilean wine industry shall be explained. In addition, the research time range is set to be 2008-2018, so what is the necessity of this research?
The introduction is rewritten, explaining the problem to be studied in a better way.
Worldwide, the wine industry has evolved rapidly [1], for winemakers, businessper-17sons and academics, regardless of their geographical area [2] [3] [4].In the same way,18information technologies have become relevant for all economic and business sectors19[5]. The above is reflected in the importance that the use of social networks is acquiring20in companies [6] and how engagement management [7], [8],[9], make a difference in a21connected and versatile world.22The production and the wine market are part of an internationally prominent23economic sector, countries such as France, Italy, Spain and Portugal are world leaders in24the wine industry [1]. Countries such as the United States, Australia, Argentina, New25Zealand, Chile, and China are also part of this list. While most of the studies of this26industry are concentrated in developed countries, there are little advances for developing27economies [10]. The Chilean wine industry experienced a dynamic growth in recent28years [11], the value of Chilean wine exports has increased from US$20 million in the29second half of the 1980s to more than US$ 1,400 million on average in the period 2005 to302007 [12]. However, it is recognized that this industry has been innovative in terms of31winemaking and trading [13].32Social media technology allows customers and companies to interact and participate33in two-way communication [14], in which both the client and company are active34participants in the generation and dissemination of content which is generated in an environment that is characterized by a network of people that are interconnected [15].36Participation in social platforms includes how consumers use, share and talk about37content related to the brand and company [16]. The first expectation of brands that use38social media marketing is the adaptation of users and contribution to the content and39interaction with the brand [17].
Today, most consumers interact with brands through social media, and brands also use social media as customer services and as a fundamental41point of contact with consumers [18].42Currently, the use of social networks [19], [20] and the interaction of its followers43with companies [8], can generate different competitive advantages compared to its44competitors [21], establishing that these relationships in social networks can be positive45or refusals from their followers or clients [22]. Companies that use their social networks46correctly can improve their internal processes, specifically in marketing [23], from this47perspective companies tend to have a better relationship with their customers [24].48The objective of this research is to identify the use and importance of engagement in49digital social networks (Facebook, Instagram and Twitter: since the creation of their offi-50cial accounts), through the analysis of 70,856 publications from the main wineries of Chile51located in the Metropolitan Region, VI Region of Libertador General Bernardo O’Higgins52and VII Region of Maule during the analysis period of a decade from 2008-2018. The53methodology used for this study is mixed qualitative and quantitative approach, where54it integrates the opinion of experts and subsequently proceeds to analyze the data sta-55tistically, implementing the calculation and analysis of engagement, in addition to a56network analysis to characterize the behavior of the vineyards based on metrics from57Facebook, Instagram, and Twitter.58Regarding the literature review, there is evidence of a lack of studies on issues59related to [10] in countries of emerging economies [25] and the null study of engagement60and social networks in the wine industry. The contribution of this research is the genera-61tion of empirical evidence for the wine industry, enhancing its competitive advantage62through social networks and engagement.63This article is structured as detailed below: starting with section 2 that addresses the64literature review of social networks, engagement and the wine industry, to continue with65section three that focuses on the methodology, where it is explained how the sampling66was conducted, the data collection procedure and respective analysis, the results are67continued, thus ending with the presentation of the conclusions and future lines of68research.
- The citation in Line44 shall follow the MDPI style.
When restructuring the introduction, line 44 no longer exists.
- In Line 58-62, the manuscript has insufficient supporting evidence for the lack of relevant research.
Lines 58-62 have been restructured.
Included are bibliographies of Román, J.J.; Cancino, C.A.; Gallizo, J.L.Exploring features and opportunities of rapid-growth wine firms in Chile.Estudios Gerenciales2017,33, 115–123.doi:https://doi.org/10.1016/j.estger.2017.02.004.
- Please reconfirm the structure in Line 63-68.
Lines 63-68 have been restructured. Since lines 64-69.
This article is structured as detailed below: starting with section 2 that addresses the64literature review of social networks, engagement and the wine industry, to continue with65section three that focuses on the methodology, where it is explained how the sampling66was conducted, the data collection procedure and respective analysis, the results are67continued, thus ending with the presentation of the discussion, the conclusions and68future lines of research.
- In Line 69-143, there are structural problems, and attention shall be given to the consistency of the article structure. For example, it only has 1.1.1 Engagement.
Lines 69-143 have been restructured, and 1.1.1 Engagement. Is replaced by a) Engagement.
- In the Materials and Methods section, there are differences in the sampling ratios for Facebook, Instagram, and Twitter—in particular, the sampling ratio for Twitter exceeding 50% of the total sample size. What’s more, to my knowledge, Instagram was launched in 2010, but the data offered was from 2008 to 2018, which made me concerned about the accuracy of data analysis.
In relation to the sample and temporality of this the information analyzed has a range of a decade 2008-2018, but the data collection is from the official creation of the account in the social network, making it clear in the introduction and abstract see table.
The instagram information was replaced (2012-2018).
We must clarify that the proportion of each social network in the sample corresponds to the publications available for each one. Indicated percentages are not sample intensities, the represent directly the proportion of publications of the networks in the sample considered in our study given data availability.
- The Results and Conclusions sections are poorly organized. It is recommended to rewrite the Results section and add a Discussion section. Furthermore, the conclusion is not robust enough, and I would like to see the differences between this research and previous research as well as its innovations.
The information on results and conclusions is reorganized.
Dividing into a discussion and conclusions section

Round 2
Reviewer 2 Report
Thank you for your improvements. Overall the article is in good form and suitable for publication after thorough proofreading.